# The effect of reproductive, hormonal, nutritional and lifestyle on breast cancer risk among black Tanzanian women: A case control study

Larry Onyango Akoko[1]*, Amonius K. Rutashobya[1¤], Evelyne W. Lutainulwa[2], Ally H. Mwanga[1], Sokoine L. Kivuyo[3]

1 Department of Surgery, Muhimbili University of Health and Allied Sciences, Dar es Salaam, Tanzania, 2 Programs department, ICAP-MSPH Tanzania, Dar es Salaam, Tanzania, 3 National Institute for Medical Research, Muhimbili Branch, Dar es Salaam, Tanzania

¤ Current address: Department of Surgery, Benjamin Mkapa Referral Hospital, Dodoma, Tanzania
* akokole12@gmail.com

## Abstract

### Purpose

This study aimed to determine the effect of reproductive, hormonal, lifestyle and nutritional factors on breast cancer development among Tanzanian black women.

### Methodology

We undertook a case-control study age-matched to ±5years in 2018 at Muhimbili National Hospital. The study recruited 105 BC patients and 190 controls giving it 80% power to detect an odds ratio of ≥2 at the alpha error of <5% for exposure with a prevalence of 30% in the control group with 95% confidence. Controls were recruited from in patients being treated for non-cancer related conditions. Information regarding hormonal, reproductive, nutritional and lifestyle risk for breast cancer and demography was collected by interviews using a pre-defined data set. Conditional multinomial logistic regression used to determine the adjusted odds ratio for variables that had significant p-value in the binomial logistic regression model with 5% allowed error at 95% confidence interval.

### Results

The study recruited 105 cases and 190 controls. Only old age at menopause had a significant risk, a 2.6 fold increase. Adolescent obesity, family history of breast cancer, cigarette smoking and alcohol intake had increased odds for breast cancer but failed to reach significant levels. The rural residency had 61% reduced odds for developing breast cancer though it failed to reach significant levels.

**Data Availability Statement:** All relevant data are within the manuscript and its Supporting Information files.

**Funding:** The author(s) received no specific funding for this work.

**Competing interests:** The authors have declared that no competing interests exist.

## Conclusion

Older age at menopause is a significant risk factor for the development of breast cancer among Tanzanian women. This study has shed light on the potential role of modifiable risk factors for breast cancer which need to be studied further for appropriate preventive strategies in similar settings.

## Background

Breast cancer (BC) is now the leading cause of cancer globally, surpassing lung cancer by contributing to 2.3million new cases and 684,679 cancer related deaths [1]. The burden of BC has geographical variability with an elevated incidence in high-income countries [2]. In spite of a somewhat low burden, it is second only to cervical cancer in many sub-Saharan African countries [3]. In Tanzania, it is estimated that BC contributes to 3,037 (7.2%) incident cancer cases and 1,303 (4.6%) cancer-related deaths [4]. This makes the control of BC an important public health priority in many countries including in Tanzania.

The global burden of cancer had recently shifted to low and middle income countries that now account for 57% of new diagnoses and 65% of all cancer related deaths [5]. About half of new BC diagnosis and more than half of its related deaths are now in LMICs [6, 7]. Transitioning countries in Africa, Asia, and America are experiencing rising incident rates [8]. The exact reason for this increase in burden is not fully understood due to lack of understanding on the local risk factors. Failure to understand the local risk factors, coupled with late presentation due to lack of screening services among other reasons, might be responsible for the rising case fatalities rate in most of sub-Saharan Africa.

Most of the known risk factors have only been fully studied in western countries but not in low income countries. Studying BC risk factors is important as non-hereditary risk factors are responsible for the vast majority of cases [9]. There is limited knowledge on how geographic variability is related to the etiologic factors that have been studied. Understanding how the hormonal (exogenous and endogenous), anthropometric and lifestyle risk factors have shaped the epidemiology of BC is vital. This will allow adoption of local strategies to address the burden of BC in these settings. This study, therefore, was designed to understand the role of the known etiologic risk factors for BC among indigenous Tanzanian women with BC. This paper presents the association between reproductive, hormonal, nutritional and lifestyle risk factors for BC among Tanzanian black women.

## Materials and methods

### Study design and setting

This was a case-control study utilizing a locally designed and pretested questionnaire designed to evaluate the potential effect of the known reproductive, hormonal, lifestyle and nutritional risk factors for the development of breast cancer (BC) among indigenous Tanzanian women. The study was conducted at Muhimbili National Hospital (MNH) in 2018 for 4 months between August and December. MNH is a publicly owned teaching and national tertiary referral hospital affiliated with Muhimbili University of Health and Allied Sciences (MUHAS). The hospital is a multi-disciplinary entity receiving all kinds of diagnoses, benign and malignant, surgical, and medical. Patients with suspected and confirmed diagnosis of BC are derived from the whole country, as it is the only hospital where comprehensive cancer services can be offered

in the country along with its sister institution (Ocean Road Cancer Institute) that offers medical and radiation therapy. The hospital also enjoys a mixture of public and private patients. This makes MNH an ideal recruitment center for both the cases and controls.

### Inclusion criteria

**Cases.** The study identified all consecutive cases suspected or diagnosed with BC in the surgical wards of the hospital. Patients with a breast lump underwent Fine Needle Aspiration Cytology (FNAC) or a core needle biopsy where FNAC was none conclusive to confirm the diagnosis. For cases that came with a confirmed diagnosis from another facility, a slide block review was carried out at our pathology laboratory. The presence of pathologist signed report with assigned histology or cytology number from MNH central pathology laboratory was considered diagnostic hence assigned as cases. Additionally, cases were indigenous Tanzanian women with no racial mix and with no previous residence in any high income country. Cases were consecutively recruited to the desired sample size from the inpatient. At the end of the recruitment, cases were found to be between 25 and 85 years of age.

**Controls.** Controls were considered to be women who were ≥20 years of age, unrelated to the cases, with no prior history of breast disease or any malignancy and no history of residency in a high income country at any one point. They were selected among patients admitted in general medical wards for non-malignancy related conditions. Once a case was identified, two suitable controls were identified age-matched to ±5 years within 48hours of identification of cases (age range of the controls was between 21 and 8 4years). A physical breast examination was carried out to rule out any breast lump: there was no mandatory requirement for mammographic evaluation to rule out subclinical breast lumps. Both cases and controls were indigenous African women who were Tanzanian nationals with no history of having stayed outside the country at any point in time.

### Study variables

A survey instrument was developed to capture the participant's sociodemographic details that included: a date of birth in years; highest level of education awarded; residency as rural is settling in the village or urban for both small and large towns; occupation as source of income for managing daily living; and marital status as categorised into two groups as the unmarried comprising of single, widowed and divorcees, and the married for those in any stable relationship as marriage or cohabiting. Risk factors collected were related to the age of onset of menarche and menopause: menarche was taken as the occurrence of a first menstrual flow, while menopause was considered when a woman reported 12 consecutive months with no menses either spontaneous or post-surgery. Twelve (12) years and 45 years was set to determine early vs delayed onset for menarche and menopause respectively age at first childbirth included live births or pregnancy loss beyond eight months with categorization as parous or nulliparous, delivery beyond 35 years was considered as late. Additional factors collected included history of breast feeding (BF), ever use of any modern contraceptives to control child birth, fertility drug use to stimulate ovulation use, any alcoholic substance use, active smoking. Any breast cancer reported by a first degree relative was also collected. A self-reported sense of obesity during adolescent life was the only nutritional component assessed.

### Study power

The study initially aimed to recruit 87 cases and 174 controls; however, due to the ease of accrual of cases, we managed to recruit 105 BC patients and 190 controls. This allowed the

study to have 80% power to detect an odds ratio of $\geq 2$ at the alpha error of $<5\%$ for exposure with a prevalence of 30% of the risk factor in the control group with 95% confidence.

### Data collection

A predesigned English questionnaire was translated to Swahili and back translated to English. This was field tested for validity and reliability on 10 patients and 10 controls. Two research assistants were trained to administer the questionnaire to the study subjects. Once patient with a breast mass was identified, preliminary screening including obtaining consent was done. Upon confirmation of histology with available signed report, those meeting the study inclusion criteria then underwent a complete data abstraction through a provider administered interview to complete the questionnaire. Concurrently, for every case recruited, an age matched control was sought from the medical ward within 48 hours of the case identification. The same questionnaire used by the cases was administered for the controls.

### Data analysis

After checking for completeness, the collected data were de-identified to keep study subjects anonymity and entered into Statistical Package for Social Scientists (SPSS) version 24.0 for further analysis. Cochran-Mantel-Hanszel (CMH) testing, stratified by the case-control pair was used to identify the risk factors for BC in a binomial logistic analysis, which provided estimates of the odds ratios (OR) and 95% confidence intervals (CI) of the risk factors. Furthermore, multinomial logistic regression was performed for variables that showed significant levels in CHM logistic regression modeling: a backward stepwise selection of variables whose p-value was $\leq 0.07$. The adjusted ORs and corresponding 95% CI's from this final model provided the comprehensive and less biased estimates of the risk factors associated with BC after adjusting for possible confounding variables and covariates.

### Ethical consideration

The study protocol was reviewed and approved by the Muhimbili University of Health and Allied Sciences Institutional Review Board. Separate permission was also obtained from MNH education, research, and consultancy bureau. Written informed consent was obtained in Swahili from all participants before enrolment. There was no monetary compensation to the participants. Private consultation rooms were used to conduct interviews to ensure privacy and confidentiality. To protect patient health information, no names were recorded, and each participant was assigned a unique study identification number. Furthermore, this study] adhered to the Helsinki declaration on studies involving human subjects.

## Results

During the study period, 105 cases and 190 age matched controls were recruited into the study, giving a case to control ratio 0f 1:1.8. Table 1 represents the demographic characteristics of the study subjects. Controls and cases were similar in all aspects except for: age where by cases were about five year's younger (p = 0.006); and there were 13.9% control group coming from urban residence (p = 0.19).

### Binomial logistic regression

Since age and residency were not uniformly distributed between cases and controls, they were added to the binomial regression model along with other known risk factors to study any association with BC risks in Table 2. Family history of breast cancer (p, 0.006), Cigarette smoking

**Table 1. Comparing sociodemographic characteristics of breast cancer patients and their age matched controls at Muhimbili National Hospital, Dar es Salaam.**

| Variable | Case | Control | P-value |
|---|---|---|---|
| **Patients mean Age** | 49.55±13.8 | 44.95±13.5 | 0.006 |
| | (25–85) | (21–84) | |
| **Education level** | | | |
| Primary and less | 82 (78.1%) | 135 (71.1%) | 0.189 |
| Secondary and above | 23 (21.9%) | 55 (28.9%) | |
| **Residence** | | | |
| Urban | 55 (52.4%) | 126 (66.3%) | 0.019 |
| Rural | 50 (47.6%) | 64 (33.7%) | |
| **Occupation** | | | |
| Formally Employed | 14 (13.3%) | 31 (16.3%) | 0.495 |
| Unemployed | 91 (86.7%) | 159 (86.7%) | |
| **Marital status** | | | |
| Unmarried | 46 (43.8%) | 65 (34.2%) | 0.103 |
| Married | 59 (56.2%) | 125 (65.8%) | |

(p, 0.004), adolescent obesity (p, 0.0007), age at menopause below 45 years (p, 0.07), and older mean age (p, 0.006) had shown significant. The rural residency was found to be protective for BC development, with a 56% reduced risk among female with rural residency (p, 0.019). The following factors had increased risk but failed to rich significant levels: alcohol intake had 43% increased risk, and null parity had 2.5 more risks.

**Table 2. Showing binary analysis results comparing cases and controls for risk factors for breast cancer development among African women in Tanzania.**

| RISK FACTOR | CASE | CONTROL | OR | P-value | 95% CI |
|---|---|---|---|---|---|
| **Family history of breast cancer** | 16 (15.2%) | 10 (5.3%) | 3.2 | 0.006 | 1.40–7.40 |
| **Adolescent obesity** | 46 (43.8%) | 53 (27.9%) | 2.0 | 0.0007 | 1.20–3.32 |
| **Cigarette Smoking** | 12 (11.4%) | 5 (2.6%) | 4.7 | 0.004 | 1.63–13.96 |
| **Alcohol intake** | 43 (41.0%) | 62 (32.6%) | 1.43 | 0.15 | 0.87–2.35 |
| **First delivery below 35 yrs.** | 96 (91.4%) | 182 (95.8%) | 0.79 | 0.79 | 0.12–4.78 |
| **Null parity** | 8 (7.6%) | 6 (3.2%) | 2.5 | 0.94 | 0.85–7.5 |
| **Contraceptive use** [a] | 56 (53.3%) | 99 (52.1%) | 1.05 | 0.84 | 0.65–1.69 |
| **Breast feeding** | 96 (98%) | 184 (99.5%) | 0.26 | 0.28 | 0.02–2.90 |
| **Infertility drug use** [b] | 14 (13.3%) | 26 (13.9%) | 0.97 | 0.93 | 0.48–1.95 |
| **Age at menarche (years)** | | | 1.13 | 0.83 | 0.36–3.58 |
| < 12 | 5 (4.7%) | 8 (4.2%) | | | |
| ≥ 12 | 100 (95.2%) | 182 (95.8%) | | | |
| **Age at menopause** [c] **(years)** | | | | | |
| < 45 | 16 (28.1%) | 10 (14.7%) | 2.26 | 0.07 | 0.93–5.49 |
| ≥ 45 | 41 (71.9%) | 58 (85.3%) | | | |
| **Mean age in years (range)** | 49.55±13.8 (25–85) | 44.95±13.5 (21–84) | 0.97 | 0.006 | 0.96–0.99 |
| **Residence** | | | | | |
| Urban | 55 (52.4%) | 126 (66.3%) | 0.56 | 0.019 | 0.343–0.91 |
| Rural | 50 (47.6%) | 64 (33.7%) | | | |

[a] Any reported use of a modern contraceptive method.

[b] Any reported modern medicine taken to boost ovulation or sustain a pregnancy.

[c] Only 57 cases and 68 controls had attained menopause at the time of the study.

## Multinomial conditional logistic regression

The following factors which had a p-value of ≤0.07 from the CHM model were brought into the multinomial logistic regression model in Table 3 below: participants mean age, family history of BC, history of smoking, age at menopause, history of obesity, and place of residence. AOR with a 95% confidence interval was then calculated in the multinomial model. Attaining menopause above 45 years of age had a 2.6-fold increased risk of developing BC and this was significant (p = 0.045, 95%CI 1.02–6.94). The presence of adolescent obesity had a 2-fold increased risk but failed to reach significant levels (p = 0.064, 95%CI 0.96–4.66). Similarly, smoking and family history of BC had 3-fold and 2-fold increased risk respectively but failed to reach significant levels (p = 0.13, 95%CI 0.74–11.55, and p = 0.153, 95%CI 0.75–7.11 respectively). The Urban residency seemed to have an insignificant, (p = 0.22, 95%CI 0.28–1.34), protective effect with a 0.61 reduced odds of developing breast cancer compared with rural residency.

## Discussion

This study evaluates the role of age, hormonal, reproductive, and lifestyle risk factors for BC among indigenous Tanzanian women. While some of the risk factors we present are the same as those in HICs, others did not prove to be relevant in our setting and hence remain controversial calling for more expanded epidemiological studies.

## Reproductive factors

Early age at menarche, null parity, older age at first childbirth, not breastfeeding, and older age at menopause are long known to be associated with an increase in BC risk in high income and transitioning countries [10]. While onset of menarche and menopause "cannot" be controlled, the majority of Tanzanian women: had first child birth when they were younger than 35 years of age and had more than one child birth. Likewise, most of these women were breastfeeding unlike in the west. These observations can explain the seemingly low prevalence of BC in this

**Table 3. Multinomial logistic regression analysis showing the strength of the association between selected variables and the risk of breast cancer among African Tanzanians.**

| RISK FACTOR | AOR | P-Value | CI (95%) |
| --- | --- | --- | --- |
| **Mean age** | 1 | 0.94 | 0.96–1.04 |
| **Family history** | | | |
| Present | 2.29 | 0.153 | 0.75–7.11 |
| Absent | 1 | | |
| **Cigarette Smoking** | | | |
| Yes | 2.92 | 0.13 | 0.74–11.55 |
| No | 1 | | |
| **Residence** | | | |
| Urban | 0.61 | 0.22 | 0.28–1.34 |
| Rural | 1 | | |
| **Adolescent obesity** | | | |
| Present | 2.09 | 0.065 | 0.95–4.6 |
| Absent | 1 | | |
| **Age at menopause** | | | |
| < 45 | 2.63 | 0.047 | 1.01–6.83 |
| ≥ 45 | 1 | | |

community compared to the west. Multi parity found in this study has been associated with a higher prevalence of breastfeeding (BF). Studies have shown that BF confers additional benefits to the woman such as prevention of endometrial and ovarian cancer, diabetes, and hypertension as well [11–13]. BF for 12 months consecutively has been shown to reduce the lifetime risk for the development of BC by 4.3%, whereas each additional parity conferred a 7% reduced risk [14]. But why such a high parity and prevalent BF did not confer prevention in Tanzanian women needs to be studied further. Is there an underlying factor that is inherent to being an indigenous African?

Although multiparity is protective for BC, new evidence suggests that it is only protective towards the hormone receptor-positive BC subtypes but increases the risk for the hormone receptor negative and triple-negative BC subtypes [15, 16]. Parous women are 2.8 times more likely than nulliparous women to develop triple-negative BC, but prolonged BF can reduce this risk [17]. The protective role of BF has not been documented in Herceptin-2 positive tumors but only limited to triple-negative and luminal tumors [18]. Given that black women are more prone to develop triple negative BC subtypes [19], Tanzanian women stand to benefit from continued practicing BF. The practice of BF has the potential to significantly reduce the Triple negative BC subtypes among black Tanzanian women [20]. Opportunities to promote BF should be protected to maintain the higher BF rates reported in this study.

We did not collect data on the molecular subtypes of BC to study this interesting association since they are not routinely collected during work up. Previous studies suggest that more than half of women with BC in Tanzania are hormone receptor negative [21]. Likewise, up to one third of patients are reported to have a triple negative BC [22]. These two findings might explain the failure of BF and other hormonal risk factors to protect local Tanzanian women. While we encourage Tanzanian women to continue BF for its overall health benefit, these observed epidemiological factors need to be investigated alongside histological subtypes of breast cancer in the indigenous African setting.

## Hormonal

The hormonal risk for the development of BC is a well-proven fact, predominantly through the Estrogen and Progesterone receptors [23]. Hormonal risk factors are associated with hormone receptor-positive BC and not the hormone receptor-negative subtypes. Oral contraceptives have not demonstrated any additional risk for BC development over the past 20 years or so [24]. However, there is a possibility of dose-response dependence. Our study only examined the question of ever use, while the answer lies in the duration of usage. Future case control studies should include data on duration of usage for hormonal contraceptive and analyze data inclusive of tumor biology to hormone receptor status.

## Lifestyle

Obesity has controversial relationship with the development of breast cancer with a reduction in risk when it occurs at teenage and a modest increase when it develops after teenage [25]. In spite of this, weight loss from the most of adult life is known to reduce the BC risk [26]. This study could not make conclusive statements on obesity since it relied on individual recall of feeling obese during the adolescent period. Health records on individuals are lacking in Tanzania even when hospital visits have taken place. Moreover, BMI is not routinely assessed in many health visit encounters in Tanzania. Establishing BMI among patients during presentation in our setting has the potential of not yielding true results given the predominant late presentation and the possible accompanied tumor related weight loss. Nevertheless, future studies should include data on actual BMI by taking actual measurements or calculating from the last

known weight. Likewise, dietary habits of these women with breast cancer need to be investigated. While establishing obesity during the index illness was made difficult by the BC associated weight loss, promoting weight loss should be given priority among women.

Even though women consuming any alcoholic drink were twice at risk for the development of BC compared to those who did not, caution is needed in interpreting the failure to reach the desired significance level. This could be due to the small sample size and this study was not powered to detect alcohol effect. It has been shown that alcohol increases mammographic density, a known risk factor for developing BC [27]. About 1–2 drinks per day have been linked with a 15–30% increased risk for BC [28]. With 41% of cases and 32% of controls reporting ever drinking, alcohol effect can be studied longitudinally to establish its exact role on BC causation in our setting. Since there are different types of drinks, local and commercial brew, these need to be taken into consideration when investigating the potential role of alcohol.

Mammary tissue is capable to uptake various tobacco carcinogens, including polycyclic aromatic hydrocarbons, aromatic amines, and N-nitrosamines which have demonstrated an in vitro capacity to induce malignancy in breast cells [29]. Thus smoking has been positively linked to the development of BC [30], especially the hormone receptor-negative subtype but not on the triple-negative [31]. This is almost similar to what we report, though it failed to reach confidence levels set for the study. Since smoking is modifiable, more studies to establish a clear link among Tanzanian women who develop BC are needed. It is important that characterization of smoking among Tanzanian women be carried out to fully understand how it relates to BC development.

There were more BC patients in our study that had urban residences than those from rural residence. This finding is similar to that from China that concluded that BC is actually higher among urban women probably due to a higher socioeconomic status compared with rural women [32]. Westernization and changing reproductive patterns accompanying urban life in China has been linked to an increase in BC incidence [33]. There are similar reports of rural-urban disparities in BC risk from India, with reproductive and central adiposity to blame [34]. The rural-urban incident disparity of BC in Tanzania needs to be confirmed and established by a larger, multicenter study.

The study being set only for patients reaching a tertiary hospital for treatment has the potential of misrepresenting the true rural-urban disparity of BC. It is known that risk factors distribution might not be the same between rural and urban dwellers, especially on modifiable ones [35]. With Tanzania eying middle income status, more women are going to migrate to urban locations and this pose a threat to increase in the incidence of BC if nothing is done to understand the pre-existing risk factors. Preventive measures needed for the two population groups might not be the same across a large country like Tanzania.

## Study limitation

Having been set at a tertiary level, the controls might not reflect the environmental influence on the studied risks since cases and controls were not matched based on regional status. We observed that controls were mostly from urban setting hence reaching a conclusion on rural vs urban became difficult. Future studies should match based on this factor too. Furthermore, with the recent improvement in hospital and cancer services in some regions, the cases might not reflect the true population of the Tanzanian population. Hence we propose a multicenter, in country case control study to fully understand the breast cancer risk factors among Tanzanian women.

Recall bias could not be ruled out, especially in older women: recalling precisely menarche and obesity seemed challenging. There was a failure to age match the cases with controls

which might fail to expose the true impact of the evaluated risk factors. Even though the sample size had the power to detect the significant odds, but the prevalence of exposure in some of the control variables did not reach the 30% needed for this study. Additionally, anthropometric measurements are needed to further evaluate their role rather than relying on reported self-feeling of obesity.

## Conclusion

We have demonstrated that older age at menopause has an increased risk for the development of BC among indigenous Tanzanian women. The remainder of the hormonal, reproductive, lifestyle and nutritional factors had non-conclusive results though they had suggested some causative association: more studies are needed to further evaluate their role. This study is important in shedding light on the plausible role of modifiable risk factors for breast cancer among indigenous African women in Sub Saharan Africa.

## Supporting information

**S1 File. English version of BC risk questionnaire.**
(PDF)

**S2 File. Swahili translation of the English questionnaire.**
(PDF)

**S1 Data. SPSS data set of the study.**
(SAV)

## Acknowledgments

We acknowledge all residents' class of 2016/2019 for their contribution in the management of breast cancer patients who were recruited in this study. Secondly, our department secretary Mrs. Agatha Haule, for her secretarial helps with the writing.

## Author Contributions

**Conceptualization:** Larry Onyango Akoko, Amonius K. Rutashobya.

**Data curation:** Amonius K. Rutashobya.

**Formal analysis:** Larry Onyango Akoko, Amonius K. Rutashobya, Ally H. Mwanga, Sokoine L. Kivuyo.

**Methodology:** Larry Onyango Akoko, Amonius K. Rutashobya, Evelyne W. Lutainulwa, Ally H. Mwanga, Sokoine L. Kivuyo.

**Project administration:** Larry Onyango Akoko.

**Supervision:** Larry Onyango Akoko.

**Validation:** Larry Onyango Akoko, Evelyne W. Lutainulwa.

**Writing – original draft:** Larry Onyango Akoko, Amonius K. Rutashobya, Evelyne W. Lutainulwa, Ally H. Mwanga, Sokoine L. Kivuyo.

**Writing – review & editing:** Evelyne W. Lutainulwa, Ally H. Mwanga, Sokoine L. Kivuyo.

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
