## [Decision Letter · Decision Letter 0]

23 Jun 2021

PONE-D-21-08470

The effect of reproductive, hormonal, nutritional and lifestyle risk on breast cancer among black Tanzanian women: A case control Study.

PLOS ONE

Dear Dr. Akoko,

Thank you for submitting your manuscript to PLOS ONE. After careful consideration, we feel that it has merit but does not fully meet PLOS ONE’s publication criteria as it currently stands. Therefore, we invite you to submit a revised version of the manuscript that addresses the points raised during the review process.

We look forward to receiving your revised manuscript.

Kind regards,

David Teye Doku

Academic Editor

PLOS ONE

Journal Requirements:

2.  In your Methods section, please provide additional information about the participant recruitment method and the demographic details of your participants. Please ensure you have provided sufficient details to replicate the analyses such as

a) a description of all inclusion/exclusion criteria that were applied to participant recruitment,

b) a statement as to whether your sample can be considered representative of a larger population, and

c) a description of how participants were recruited.

3. Thank you for stating in the text of your manuscript "Written informed consent was obtained in Swahili from all participants before enrolment. " Please also add this information to your ethics statement in the online submission form.

4. Please include additional information regarding the survey or questionnaire used in the study and ensure that you have provided sufficient details that others could replicate the analyses. For instance, if you developed the survey or questionnaire as part of this study and it is not under a copyright more restrictive than CC-BY, please include a copy, in both the original language and English, as Supporting Information. If the questionnaire is published, please provide a citation to the (1) questionnaire and/or (2) original publication associated with the questionnaire.

Additional Editor Comments (if provided):

Reviewers' comments:

Reviewer's Responses to Questions

**Comments to the Author**

1. Is the manuscript technically sound, and do the data support the conclusions?

Reviewer #1: Partly

Reviewer #2: Partly

2. Has the statistical analysis been performed appropriately and rigorously? 

Reviewer #1: No

Reviewer #2: Yes

3. Have the authors made all data underlying the findings in their manuscript fully available?

Reviewer #1: Yes

Reviewer #2: Yes

4. Is the manuscript presented in an intelligible fashion and written in standard English?

Reviewer #1: No

Reviewer #2: Yes

5. Review Comments to the Author

Reviewer #1: Review of the Manuscript entitled :-

“The effect of reproductive, hormonal, nutritional and lifestyle risk on breast cancer among black Tanzanian women: A case control Study”

Abstract:

In the Purpose: The word “Life style” which was mentioned in the main Title is not mentioned here.

In Results:

• In line 40-41 the word “Smoking” was unnecessary repeated twice (smoking ¬– cigarette smoking)

• Line 42 , Nulliparity is missed here.

Manuscript:

• Line 1: in Full title: Life style is missed

• Line 53: in Key words: Tanzanian women could be added

• Line 96: in Study subjects: age range of the studied sample should be added.

• Lines 189 & 191: values of P & CI to be matched with the table

• Line 214-216: if these results are of the present study please mention if not add a reference

• Line 267: should be explained more clearly e.g.” In rural areas the % of cases was more than that of controls and the difference was significant, (47.6% cases vs 33.7% controls, p 0.019)”

• In the whole study: The study neglected to evaluate the potential effect of nutritional risk factors for the development of breast cancer throughout the results as well as the discussion although it was mentioned in the methodology.

• References: the last resent reference is from 2019, it would be better to add some more recent ones.

General Comments on Statistics

1. As shown in the results, there was a significant difference between the mean ages of cases and controls which denote that controls were not properly age matched with cases, (P= 0.006).

2. Regarding Residence we can notice contradictory results: while in Table-1-, for rural residencies there was a significant risk for BC (p=0.019), in Table-2 & 3, rural residencies have a protective effect regarding BC, (P=0.019, OR 0.56 & P=0.22, AOR 0.61 respectively).

3. Regarding Age at menopause:

a. In Table -2- it was mentioned that age at menopause < 45 is a sig. risk ( P=0.07, OR 2.26), while in Table -3- age ≥ 45 had 2.6 fold increase risk of developing BC and is significant (P= 0.047, AOR 2.63).

b. Changing the order of the risk factor (<45 and ≥ 45) in the statistical analysis (Tables 2 &3) led to such contradictory results.

Reviewer #2: The submitted manuscript entitled (The effect of reproductive, hormonal, nutritional and lifestyle risk on breast cancer among black Tanzanian women: A case control Study) aimed to define some environmental factors that may play a role in the breast cancer such as hormones and nutrients according to the lifestyle.

The manuscript is a case study and well-presented and is a representative for the factors under investigation.

There are few minor comments:

1- Line 159: as mentioned that cases were significantly older and came from rural residences. However, the results showing that the number and percentage of urban is more than the rural in both control and cases.

2- For the age parameter, it is difficult to consider the average as a realistic so as I understood that it is analyzed later in details. However, it is still confusing about the age between 12 and 45.

3- The discussion part requires some modification to be more solid and informative.

4- The lifestyle part of discussion especially the part of residency as the results in table 2 showed that the more cases and controls came from urban in contrast to the discussion part.

6. PLOS authors have the option to publish the peer review history of their article (what does this mean?). If published, this will include your full peer review and any attached files.

Reviewer #1: **Yes: **Fatma Ahmed El Sayed Shaaban

Reviewer #2: **Yes: **Ahmed K Elsayed

---

## [Author Response · Author response to Decision Letter 0]

4 Aug 2021

20th July, 2021

Dear editor

Thanks again for giving us the opportunity to critically scrutinize our paper after reviewer’s comments and those of your own. I have incorporated the response to all the comments as was appropriately demanded. Secondly, I have adhered to the journal stylistic demands for the body and tittle page. Likewise, the supporting section has been added accordingly including both versions of the questionnaire used in this study. 

a) You suggested that we add participant’s recruitment methods and their demographics in the methodology section which we have now made. We have a subsection that is addressing only recruitment of cases and controls detailing how the recruitment was made with detailed inclusion and exclusion criteria

b) On whether our sample size can be considered representative of a larger population yes and no. Most of the breast cancer patients in the country that make it to a treatment facility will end at our hospital. However, there is a possibility that there are patients who will succumb before reaching us. Likewise, medical patients might not be drawn for the entire country since their conditions are treatable in many other facilities across the country. This had the potential of bringing a mismatch between cases and controls on geographical risk if any exists. We have added this to the discussion “Para 1 line 212-214 (Since breast cancer is known to be a heterogeneous disease, the sample of women studied can only serve a snapshot of Tanzanian women but might not be generalized for all BC patients)”

c) We have added the consent statement “Written informed consent was obtained in Swahili from all participants before enrolment." To the submission portal

d) We have attached both versions of the survey as S1 Files 1 – 2

e) The caption for the supporting files has been added right after the references.

f) In addressing first issue as to whether the manuscript is technically sound, we have changed significantly issues that worried the reviewers and we now believe it is sound and that our conclusion is drawn from the results. Secondly, we have addressed the statistical issue that worried reviewer 1 adequately and I hope Prof.(Ms.) Fatma will be satisfied now. 

g) We have also used the latest GLOBOCAN statistics publication of 2020 and added four more references to enrich our discussion section.

Reviewer #1 comments with responses 

“The effect of reproductive, hormonal, nutritional and lifestyle risk on breast cancer among black Tanzanian women: A case control Study” I had actually changed my submission tittle to read the” The effect of reproductive, hormonal and nutritional risk on breast cancer among Black Tanzanian women: A case control Study. I had considered lifestyle to be covered under nutritional factors. But I have now added the lifestyle in the tittle. 

Abstract

• In the Purpose: The word “Life style” which was mentioned in the main Title is not mentioned here. So in the tittle that was on the main document I did not include the word “Life style. But now it has been added.”

• In Results:

• In line 40-41 the word “Smoking” was unnecessary repeated twice (smoking ¬– cigarette smoking): the first smoking has been removed

• Line 42, Null parity is missed here. Nulliparity was not brought to the multinomial analysis. Null parity was a rare event among Tanzanian women hence the numbers were not worth making conclusion on. 

Manuscript

• Line 1: in Full title: Life style is missed. I thought of shortening the tittle as explained in above. But I have inserted the ,’lifestyle’

• Line 53: in Key words: Tanzanian women could be added. This has been added

• Line 96: in Study subjects: age range of the studied sample should be added. I have inserted the age ranges for the cases and that of the cohort in line……

• Lines 189 & 191: values of P & CI to be matched with the table. These have been added

• Line 214-216: if these results are of the present study please mention if not add a reference. The reference has been added. Actually it was assumed to be covered in the preceding line. Reference …. Actually was referring to the statement, it was a bad referencing. 

• Line 267: should be explained more clearly e.g.” In rural areas the % of cases was more than that of controls and the difference was significant, (47.6% cases vs 33.7% controls, p 0.019)”. This was actually an overlook of the actual data that we presented in the table. Most of the cases had urban residence compared to rural residence. This could partly be explained that that hospital was itself located in an urban setting, and whether all women with BC in Tanzania make it to the hospital remains unknown. 

• In the whole study: The study neglected to evaluate the potential effect of nutritional risk factors for the development of breast cancer throughout the results as well as the discussion although it was mentioned in the methodology. I must admit that we only looked at the consequence of nutrition, that is obesity at childhood, and not the dietary issues as we thought it would have been too wide to cover. In the discussion, we therefore combined the obesity with other lifestyles issues as one. 

• References: the last resent reference is from 2019, it would be better to add some more recent ones. BC risk factor might be an old topic in the developed world but which has received little attention in the LMICs. However, I have updated the GLOBOCAN reference to bring the most recent one of 2020.

General Comments on Statistics

1. As shown in the results, there was a significant difference between the mean ages of cases and controls which denote that controls were not properly age matched with cases, (P= 0.006). I take it as a failure in matching. As we indicated in the methodology, the ±5 was the same difference at analysis which turned significant. Both cases and controls were under 50 years of age which was somewhat a success in the matching. But we admit this in the limitations of our study, now added.

2. Regarding Residence we can notice contradictory results: while in Table-1-, for rural residencies there was a significant risk for BC (p=0.019), in Table-2 & 3, rural residencies have a protective effect regarding BC, (P=0.019, OR 0.56 & P=0.22, AOR 0.61 respectively). I have admitted in the comment on line 264 above that it was an oversight. Actually most patients had urban residence and so were the controls. So I have changed the discussion as well to reflect this. 

3. Regarding Age at menopause:

a. In Table -2- it was mentioned that age at menopause < 45 is a sig. risk ( P=0.07, OR 2.26), while in Table -3- age ≥ 45 had 2.6 fold increase risk of developing BC and is significant (P= 0.047, AOR 2.63).

b. Changing the order of the risk factor (<45 and ≥ 45) in the statistical analysis (Tables 2 &3) led to such contradictory results. Totally agree, there was an error assigning the symbols when making the table. I have rerun the SPSS and the changes made and the previous interpretation is correct. 

Reviewer #2

The submitted manuscript entitled (The effect of reproductive, hormonal, nutritional and lifestyle risk on breast cancer among black Tanzanian women: A case control Study) aimed to define some environmental factors that may play a role in the breast cancer such as hormones and nutrients according to the lifestyle.

The manuscript is a case study and well-presented and is a representative for the factors under investigation. Thanks for this positive and encouraging observation made on our paper. 

There are few minor comments:

1. Line 159: as mentioned that cases were significantly older and came from rural residences. However, the results showing that the number and percentage of urban is more than the rural in both control and cases. I have regrettably noted this serious error and appropriate correction has been made. 

2. For the age parameter, it is difficult to consider the average as a realistic so as I understood that it is analyzed later in details. However, it is still confusing about the age between 12 and 45. Age at menarche is the one which was taken at 12 as cut off for early or late menarche, while 45 was taken as cut off for early or late menopause. 

3. The discussion part requires some modification to be more solid and informative.

4. The lifestyle part of discussion especially the part of residency as the results in table 2 showed that the more cases and controls came from urban in contrast to the discussion part. Addressed taken care of as in your first comment above 

Larry Akoko

Corresponding author

Senior Lecturer

Department of surgery

Muhimbili University of Health and Allied sciences

---

## [Decision Letter · Decision Letter 1]

9 Sep 2021

PONE-D-21-08470R1The effect of reproductive, hormonal, nutritional and lifestyle risk on breast cancer among black Tanzanian women: A case control Study.PLOS ONE

Dear Dr. Akoko,

Thank you for submitting your manuscript to PLOS ONE. After careful consideration, we feel that it has merit but does not fully meet PLOS ONE’s publication criteria as it currently stands. Therefore, we invite you to submit a revised version of the manuscript that addresses the points raised during the review process.

Kindly address the issue raised by reviewer before final decision on your manuscript can be made. Please ensure that your decision is justified on PLOS ONE’s publication criteria and not, for example, on novelty or perceived impact.

We look forward to receiving your revised manuscript.

Kind regards,

David Teye Doku

Academic Editor

PLOS ONE

Journal Requirements:

Reviewers' comments:

Reviewer's Responses to Questions

**Comments to the Author**

1. If the authors have adequately addressed your comments raised in a previous round of review and you feel that this manuscript is now acceptable for publication, you may indicate that here to bypass the “Comments to the Author” section, enter your conflict of interest statement in the “Confidential to Editor” section, and submit your "Accept" recommendation.

Reviewer #1: All comments have been addressed

Reviewer #2: All comments have been addressed

2. Is the manuscript technically sound, and do the data support the conclusions?

Reviewer #1: Yes

Reviewer #2: Yes

3. Has the statistical analysis been performed appropriately and rigorously? 

Reviewer #1: Yes

Reviewer #2: Yes

4. Have the authors made all data underlying the findings in their manuscript fully available?

Reviewer #1: Yes

Reviewer #2: Yes

5. Is the manuscript presented in an intelligible fashion and written in standard English?

Reviewer #1: Yes

Reviewer #2: Yes

6. Review Comments to the Author

Reviewer #1: Line 182-184, "Controls and cases were similar in all aspects except for age where by cases were about five years younger (p=0.006), and there were 13.9% control group coming from urban residence (p=0.19)

- cases were younger OR older by 5 years ( for cases mean age = 49.55 while for controls it was 44.95)

- The % of control coming from urban residence is ( 66.3%) & not 13.9%, (p=0.019).

Reviewer #2: The author covered all the comments in the revised version. It sounds good from the technical view and english writing.

7. PLOS authors have the option to publish the peer review history of their article (what does this mean?). If published, this will include your full peer review and any attached files.

Reviewer #1: **Yes: **Prof. Fatma Ahmed El-Sayed Shaaban

Reviewer #2: **Yes: **Ahmed Kamel Elsayed

---

## [Author Response · Author response to Decision Letter 1]

23 Sep 2021

No comment from reviewrs are recived this round

---

## [Editor Report · Decision Letter 2]

19 Jan 2022

The effect of reproductive, hormonal, nutritional and lifestyle on breast cancer risk among Black Tanzanian women: A case control Study.

PONE-D-21-08470R2

Dear Dr. Akoko,

We’re pleased to inform you that your manuscript has been judged scientifically suitable for publication and will be formally accepted for publication once it meets all outstanding technical requirements.

Kind regards,

David Teye Doku

Academic Editor

PLOS ONE

Additional Editor Comments (optional):

None.
---

## [Editor Report · Acceptance letter]

28 Jan 2022

PONE-D-21-08470R2 

The effect of reproductive, hormonal, nutritional and lifestyle on breast cancer risk among Black Tanzanian women: A case control Study. 

Dear Dr. Akoko:

I'm pleased to inform you that your manuscript has been deemed suitable for publication in PLOS ONE. Congratulations! Your manuscript is now with our production department. 

Kind regards, 

on behalf of

Dr. David Teye Doku 

Academic Editor

PLOS ONE